# Tribbles Pseudokinase 3 Regulation and Contribution to Cancer

**DOI:** 10.3390/cancers13081822

**Published:** 2021-04-11

**Authors:** Bojana Stefanovska, Fabrice André, Olivia Fromigué

**Affiliations:** 1Inserm, UMR981, F-94805 Villejuif, France; bojana.stefanovska@gustaveroussy.fr (B.S.); fabrice.andre@gustaveroussy.fr (F.A.); 2Gustave Roussy, F-94805 Villejuif, France; 3Orsay, Université Paris Saclay, F-91400 Gif-sur-Yvette, France; 4Department of Medical Oncology, Gustave Roussy, F-94805 Villejuif, France

**Keywords:** pseudokinase, mTOR, Akt, rapamycin, ER stress, tumor suppressor, oncogene, RNA splicing, FK506 binding protein, signaling pathway

## Abstract

**Simple Summary:**

Accumulating evidence supports a key function for Tribbles proteins in oncogenesis, both in leukemia and solid tumors. However, the exact role of these proteins is hard to define since in a context-dependent manner they can function as both oncogenes and tumor suppressors. Their complex role arises from the capacity to interact with a wide range of target molecules thereby acting as molecular scaffolds and signaling regulators of multiple pathways. This review focuses on one particular Tribbles family member, namely, TRIB3, addressing its gene and protein expression, as well as its role in cancer development and progression.

**Abstract:**

The first Tribbles protein was identified as critical for the coordination of morphogenesis in Drosophila melanogaster. Three mammalian homologs were subsequently identified, with a structure similar to classic serine/threonine kinases, but lacking crucial amino acids for the catalytic activity. Thereby, the very weak ATP affinity classifies TRIB proteins as pseudokinases. In this review, we provide an overview of the regulation of *TRIB3* gene expression at both transcriptional and post-translational levels. Despite the absence of kinase activity, TRIB3 interferes with a broad range of cellular processes through protein–protein interactions. In fact, TRIB3 acts as an adaptor/scaffold protein for many other proteins such as kinase-dependent proteins, transcription factors, ubiquitin ligases, or even components of the spliceosome machinery. We then state the contribution of TRIB3 to cancer development, progression, and metastasis. TRIB3 dysregulation can be associated with good or bad prognosis. Indeed, as TRIB3 interacts with and regulates the activity of many key signaling components, it can act as a tumor-suppressor or oncogene in a context-dependent manner.

## 1. Introduction

The Tribbles protein was firstly identified because of its critical role in coordinating morphogenesis in *Drosophila melanogaster* (fruit fly) [1,2,3]. Tribbles-deficient embryos enter mitosis early in the mesodermal cells that impairs morphogenetic movement of gastrulation, when on the contrary, Tribbles overexpression greatly reduces the number of cells per wing. In mammals, three Tribbles protein homologs exhibit a high degree of amino acid similarity: TRIB1 (alias C8fW or SKIP1), TRIB2 (C5fW or SKIP2), and TRIB3 (NIPK, SINK, or SKIP3). The kinase-like domain is highly conserved through evolution suggesting an important role in the function for the Tribbles family of proteins [4]. As suggested in *Drosophila*, a relationship with cell cycle or cell survival in mammalian cells was also reported. TRIBs exhibit either unique or overlapping functions (reviewed in [5]). Such similarities and discrepancies in a pathological context upgrade TRIBs as prognostic or predictive biomarkers of disease and putative valuable therapeutic targets.

In humans, TRIB3 expression is ubiquitous. Human tissue profiling indicates the thyroid gland, bone marrow, and peripheral blood leukocytes as sites of high TRIB3 mRNA expression [6]. The Human Protein Atlas project also assessed the expression level of TRIB3 mRNA and the abundance of TRIB3 protein in a variety of tissues (www.proteinatlas.org accessed on 6 January 2021). Clinical studies report an increased TRIB3 expression in tumor tissue as compared to normal tissue and a correlation with poor prognosis. In fact, colorectal cancer patients with high TRIB3 protein and mRNA expression levels are likely to experience a recurrence of the disease and displayed poorer overall survival [7]. In the same way, in renal cell carcinoma [8], gastric cancer [9], oral tongue squamous cell carcinoma [10], and non-small-cell lung carcinoma [11], the upregulated TRIB3 expression correlates with tumor stage, lymph node metastasis, disease recurrence, and thus with unfavorable prognosis. Nevertheless, despite the association of TRIB3 upregulation with bad prognosis, it can act as a tumor-suppressor or oncogene in a context-dependent manner.

## 2. *TRIB3* Gene and Transcripts

The first approved name of the human *TRIB3* gene was “Tribbles homolog 3 (Drosophila)”, recalling the *Drosophila melanogaster* homolog, but in 2016, the HUGO Gene Nomenclature Committee reconsidered it to “Tribbles pseudokinase 3”, putting in evidence its function. The human *TRIB3* gene (gene ID: 57761) is located in chromosome 20 at the position 20p13 and spans a region of 16.9 kb. Thanks to alternative transcription start sites (TSS) and alternative splicing, it generates four transcripts (Figure 1).

The Ensembl genome browser annotates TRIB3-201 transcript as the principal transcript, and TRIB3-202 as the alternative transcript (Table 1). The protein encoded by TRIB3-202 transcript harbors 27 additional amino acids at the N-term in comparison to the one encoded by TRIB3-201. However, there is no experimental evidence indicating that this difference in sequence translates in functional diversity in the two proteins. Furthermore, the existence of protein isoforms related to TRIB3-203 and TRIB3-204 transcripts (Transcript Support Levels (TSL) of 5 and 3, respectively) has never been proven empirically, resulting in TRIB3-201 being the only transcript that encodes the TRIB3 protein.

Alternative transcription start sites add complexity to the human *TRIB3* gene expression. In silico analysis of the 5′-Untranslated Transcribed Region (5′UTR) suggests three putative alternative promoters with assigned transcripts (Figure 2). One putative promoter could generate a very short isoform of TRIB3 (encoding a protein truncated to 60 aa) for which no matching information is available to our knowledge. In contrast, the other two promoters allow the transcription of up to ten putative transcripts that finally match the TRIB3-201 principal transcript and the TRIB3-202 alternative transcript. Since all of the transcriptional differences of TRIB3 isoforms condensed in the first exon that is not protein coding, all of the TRIB3 generated proteins are identical.

## 3. TRIB3 Protein Structure

The 3D structure for TRIB3 has not been established yet, but insights from the crystal structure of its homolog TRIB1 suggest that Tribbles proteins fold into the canonical conformation of protein kinases [13]. Indeed, the TRIB3 protein presents a structure similar to a classic serine-threonine protein kinase with N- and C-terminal domains flanking a kinase-like domain in the center of the protein (Figure 3).

The kinase-like domain is highly conserved through evolution suggesting an important role in the function for the Tribbles family of proteins [4]. However, even if the catalytic core contains the classic substrate-binding domain, it presents several substitutions of amino acids that are crucial for the catalytic activity of conventional kinases (Figure 3). The most evident differences are (1) the lack of the highly conserved Asp-Phe-Gly (DFG) motif which is required for the binding of Mg^2+^ and subsequent activation of canonical kinases; and (2) the lack of the glycine-rich loop (GXGXXG) usually involved in ATP-binding. In vitro experiments confirmed no phospho-transferase activity in presence of several different classical Ser/Thr kinase substrates [14]. However, even though it has been demonstrated that TRIB3 can very weakly auto-phosphorylate under particular in vitro conditions, independently of divalent cations, cellular substrates of TRIB3 have not been identified yet [15]. Based on all these characteristics, TRIB3 is classified as a pseudokinase.

The *N*-terminal domain contains a large amount of proline (P), glutamic acid (E), serine (S), and threonine (T) residues. Such PEST regions are involved in Ubiquitin-mediated degradation of a variety of substrates, thus controlling the half-life of target proteins [16,17]. A nuclear localization signal (NLS) sequence allows TRIB3 to localize to the nucleus where it interacts with various transcription factors such as ATF4, CHOP, C/EBPβ, PPARγ, p65/RelA, or SMAD3. The C-terminal domain of TRIB3 contains two conserved sequences: Mitogen Activated Protein Kinase Kinase (MEK1/MAPKK) binding motif and Constitutive Photomorphogenesis Protein 1 Homolog (COP1) binding motif recognized by this subclass of E3 ubiquitin ligase.

## 4. Regulation of TRIB3 Expression

The best-described modalities of TRIB3 regulation up to date are through several forms of cellular stress. Evidence in the literature suggest that endoplasmic reticulum (ER) stress [18,19], oxidative stress, hypoxia, essential amino acid deficiency [20], oversupply or lack of glucose [21,22], or free fatty acids excess [23] contribute to the activation of the *TRIB3* promoter and lead to TRIB3 upregulation. In addition, post-translational modifications of TRIB3 contribute to the regulation of protein location, stability, confirmation, and activity.

### 4.1. Transcriptional Regulation

A recently published, in silico analysis of the *TRIB3* promoter region indicates six unique putative promoters but three of them with no transcript assigned (Figure 2). Further investigations of the region spanning the promoters #2 and #3 detected binding motifs for a large amount of transcription factors (Figure 4).

Several of these transcriptional factors effectively trigger TRIB3 expression levels in different contexts. The best described activator is ATF4 (Activation Transcription Factor 4) that is recruited to TRIB3 promoter thanks to a basic leucine zipper domain (bZIP) c/EBP-ATF response element [24]. In conditions of severe or prolonged stress, ATF4 forms a heterodimer with CHOP (C/EBP homologous protein) that promotes cell death via the induction of several pro-apoptotic genes and the suppression of the anti-apoptotic Bcl-2 proteins [25,26]. The transcriptional coactivator PGC-1α (Peroxisome proliferator-activated receptor gamma coactivator-1 alpha) also promotes TRIB3 expression which leads to induced insulin resistance in liver cells [27]. The cell-type specific transcription factor NFATc1 (Nuclear factor of activated T-cells, cytoplasmic 1) promotes TRIB3 expression in vascular smooth muscle cells upon phenamil treatment resulting in an attenuation of pulmonary artery hypertension in rats [28]. Another example is FoxO1 (Forkhead box protein O1), which according to the cellular context can either promote or repress TRIB3 expression. Hence, it controls the insulin sensitivity in hepatocytes by inhibiting the expression of TRIB3, whereas it promotes cell death in neurons under NGF deprivation by inducing the expression of TRIB3 [29,30].

Even though there is extensive literature describing the activation of TRIB3 in several stress conditions, little is known about its repression. In a recent publication from our group we demonstrated that the transcriptional co-repressor GCF2 (GC-Binding Factor 2) alias LRRFIP1 (Leucine Rich Repeat (In FLII) Interacting Protein 1) participates in *TRIB3* downregulation in cancer cells incubated with rapamycin or its derivatives (rapalogs) [12]. The canonical target of rapalogs is mTOR (mechanistic target of rapamycin). They bind with high affinity to the intracellular receptor FKBP12, a member of the FK506-binding protein (FKBP) family [31], and the resulting complex binds to the FKBP12-Rapamycin Binding (FRB) domain of mTOR that leads to its allosteric inhibition. Interestingly, rapamycin can also bind FKBP25 with high affinity [32] that in turn regulates GCF2 recruitment to the promoter of TRIB3 [12]. Moreover, the consensus-binding motif for GCF2 overlaps the RNA polymerase II binding motif (TF2B) near the TSS in the TRIB3 promoter, probably interfering with the recruitment of RNA polymerase II, and the initiation of transcription.

### 4.2. Post-Translational Regulation

Several enzymatic modifications occurring at a post-translational stage may regulate TRIB3 function or location. Thus, high-throughput proteomic analyses followed by mass-spectrometry confirmed the presence of seven phosphorylation sites on TRIB3 protein (Figure 5). However, to date, there is no information about the associated kinases or the functional significance of such phosphorylation. The histone-Lysine N-Methyltransferase SMYD1 (SET and MYND Domain Containing 1) triggers TRIB3 methylation at K16 in the N-term region (Figure 5). Following this methylation, TRIB3 acts as a co-repressor to SMYD1 and inhibits its transcription [33]. The acetylation of TRIB3 by PCAF (P300/CBP-Associated Factor) results in increased expression of TRIB3 [33,34]. A yeast two-hybrid screen identified an interaction of TRIB3 with the E3 ubiquitin ligase Seven in absentia homolog 1 (SIAH1) in mammalian cells, and demonstrated that it promotes proteasome-dependent degradation [35].

Finally, upon apoptosis-inducing treatment (TNFα/CHX or anti-Fas antibody), TRIB3 is cleaved by endoproteases of the caspases family at a site located 20 aa from the C-term [36]. This cleavage leads to accelerated cell death by emphasizing pro-caspase-3 and -7 activation. In contrast, upon ER stress-inducing treatment, TRIB3 is no longer cleaved. The pro-caspase-3 undergoes nuclear translocation leading to its non-activation and prevention of apoptosis induction [36].

## 5. TRIB3 as A Scaffold for Diverse Signaling Proteins

TRIB3 interferes with a broad range of cellular processes through non-catalytic mechanisms. Instead, it takes part in pleiotropic signaling network thanks to diverse and varied protein–protein interactions. Known TRIB3 partners are kinase-dependent proteins, transcription factors, ubiquitin ligases, spliceosome machinery components, as well as many other proteins. Since TRIB3 lacks an active catalytic site, it contributes to several processes as a scaffold protein or an adaptor. In fact, TRIB3 generally regulates other proteins by altering their subcellular localization or impeding the interactions with their enzymatic partners. The outcome of these regulations is strictly context dependent since TRIB3 can act both as activator and as repressor.

### 5.1. TRIB3 Regulation of Kinase-Dependent Proteins

Thanks to its C-term MEK1-binding domain, TRIB3 is able to deactivate multiple mitogen-activated protein kinases (MAPKs). In fact, the best-described function of TRIB3 is the negative regulation of Akt (thymoma viral proto-oncogene/protein kinase B) activity (Figure 6). The PI3K/Akt signal transduction pathway is pivotal for cell response to various extracellular stimuli. Active Akt plays a pro-survival role thanks to its multifunctional downstream signaling nodes. However, its effective activation requires phosphorylation on two key residues, namely, Thr308 and Ser473. Firstly, PI3K (phosphoinositide-3-kinase) activates PDK1 (Pyruvate Dehydrogenase Kinase 1), which phosphorylates AKT at Thr308. Then, for optimal activation, mTORC2 phosphorylates Akt on Ser437 [37]. Several evidence suggest that TRIB3 disrupts the phosphorylation at both Thr308 and Ser473 residues through a direct interaction with Akt [38]. In addition, TRIB3 can inhibit the Akt phosphorylation at Ser473 by binding to the mTORC2 component, RICTOR [39]. The TRIB3-mediated Akt inhibition does not inhibit all Akt substrates in the same way, which is an additional testament for its context-dependent regulation. For instance, FoxO and BAD phosphorylation is impaired, whereas GSK3β and PRAS40 phosphorylation remains intact.

In parallel to the modulation of the PI3K/Akt signaling pathway, TRIB3 binds also directly to extracellular signal-regulated kinase (ERK) and c-Jun N-terminal kinase (JNK), and dose-dependently regulates their phosphorylation state, leading to the modulation of cell proliferation and migration of lung adenocarcinoma cells, for example [40]. It should be noted that TRIB3 does not affect the levels of p-P38 MAPK. Similar effects of TRIB3 onto MAPK-dependent pro-survival and pro-migratory effects were reported in renal cell carcinoma [8], or breast cancer, completed with interactions with TGFβ pathway [41].

### 5.2. TRIB3 Regulation of Transcription Factors

In addition to the interaction with several protein kinases, TRIB3 binds and regulates the expression level of different transcription factors. In this manner, it contributes to the modulation of multiple cell behaviors that have whole-body impact. Through a COP1 (constitutive photomorphogenic protein 1) conserved binding motif present at the C-terminal domain, TRIB3 can recruit the E3 ubiquitin ligase COP1 and favor ubiquitination and proteasomal degradation of transcription factors [42,43,44]. For example, TRIB3 interacts with PPARα (peroxisome proliferator-activated receptor alpha) and promotes its ubiquitin-dependent degradation [45]. This favors the survival of leukemic cells and meditates the progression of acute myeloid leukemia (AML). Furthermore, it participates to a tight control of adipogenesis by interacting with PPARγ (peroxisome proliferator-activated receptor gamma), leading to the downregulation of its transcriptional activities. This negative regulation of PPARγ inhibits adipocyte differentiation [42].

TRIB3 regulates inflammatory signaling through NF-kB (nuclear factor-kappa B). However, the relationship between TRIB3 and NF-kB is quite complex. In certain instances, TRIB3 increases NF-kB activity, and leads, for example, to the induction of apoptosis in pancreatic β cells [46]. In contrast, in esophageal cancer cells, TRIB3 inhibits NF-kB signaling by interacting with RelA, a member of NF-kB family [47]. Moreover, TRIB3 can indirectly inhibit NF-kB as the downstream effector of the ATF4-CHOP pathway [48,49].

ATF4 (activating transcription factor-4) binds to and activates the promoter of chaperone encoding genes that help to restore cellular homeostasis. Under sustained stress, ATF4 also binds to and activates the promoter of CHOP (CCAAT-enhancer-binding protein homologous protein), which in turn boosts endoplasmic reticulum (ER) stress-induced apoptotic cell death. Furthermore, CHOP can form heterodimers with ATF4, C/EBPβ (CCAAT/enhancer binding protein), and C/EBPγ, then binds to the promoter of TRIB3 and contributes to its positive regulation [19]. In turn, TRIB3 inhibits the transcriptional activity of ATF4 and CHOP by binding directly their transactivation domains without affecting their DNA binding ability [18,50]. In this manner, a negative feedback-loop is established where TRIB3 represses its own promoter and fine-tunes its own expression. In conclusion, we could imagine TRIB3 as a sensor of ER-stress induced cell death. For instance, when the stress is mild, ATF4/CHOP-containing complexes induce TRIB3 expression, but the TRIB3-dependent negative-feedback loop inhibits their transcriptional activity that finally promotes cell survival. However, when the stress is severe, sustained TRIB3 overexpression leads to cell death.

In colorectal cancer cells, TRIB3 interacts with TCF4 but also with β-catenin in a dose-dependent manner [51]. This transcriptional complex induces the expression of genes related to Wnt signaling pathway and cancer stemness. Moreover, activated β-catenin increased expression of TRIB3, indicating a positive-feedback loop. In liver cancer cells, an increased expression level of TRIB3 correlates with increased expression levels of PPARγ, ATF4, ATF5, eIF2α, CHOP, and several other transcription factors related to activation of Wnt/β-catenin signaling pathway [52].

### 5.3. TRIB3 Regulation of Pre-mRNA Splicing

In the context of cancer cell resistance to rapamycin, a non-targeted liquid chromatography tandem mass chromatography (LC-MS/MS) proteomics combined to functional annotation clusters was conducted to complete the TRIB3 protein interaction network [12]. Surprisingly, in addition to the well-known TRIB3 interactors described above, the majority of the identified partners were related to the pre-mRNA splicing process. More specifically, TRIB3 interacts with core spliceosomal proteins that form the PRP19 and U2 small nuclear ribonucleoproteins (U2 snRNP) complexes, and thus is involved in pre-mRNA splicing modulation. Rapamycin treatment leads to impairment of the pre-mRNA splicing efficacy by downregulating TRIB3 expression in an mTOR independent manner [12].

Interestingly, a recent study in *Saccharomyces cerevisiae* reported that rapamycin triggers the accumulation of unusual introns as stabilized linear non-coding RNAs, associated with components of the spliceosome [53]. This accumulation of introns interferes with key biological functions in yeast, like, for instance, cell survival [54], whereas intron retention (IR) in higher eukaryotic organisms is coupled to non-sense-mediated mRNA decay (NMD) and regulates gene expression. Evidence in the literature suggest that IR can also regulate RNA stability and translation, as well as the generation of distinct protein isoforms [55]. In mammalian cells, pharmacological mTOR inhibition can lead to intron retention in lipogenic genes. In this context, IR impacts the stability of the mRNA by triggering NMD and disrupts the lipid metabolism thereby affecting cellular growth [56].

## 6. TRIB3 and Cancer

Since TRIB3 is a negative regulator of key processes and signaling pathways, it has a critical role in several pathologies. For instance, the overexpression of TRIB3 is involved in several metabolic dysfunctions such as diabetes type II and cardiovascular diseases [57]. In addition, considerable number of studies suggest that TRIB3 dysregulation plays a role in cancer development, progression, and metastasis.

### 6.1. Genetic Alterations and Expression

According to the Catalogue Of Somatic Mutations In Cancer (COSMIC) database, several somatic mutations are present in the *TRIB3* gene (Figure 7A). However, in the current version of The Cancer Genome Atlas Research Network (TCGA) that comprises cancer samples for 33 different cancer types (last access January 2021) *TRIB3* is mutated in 1.28% of cancer samples (Figure 7B). 

In the context of genetic alterations, two different studies reported a copy number gain for the *TRIB3* locus in 16% and 40% of patients [58,59]. As a result, the colorectal, liver, and lung cancer exhibit higher TRIB3 mRNA level as compared with non-tumor sample. In accordance, data from TCGA confirm that TRIB3 mRNA expression is higher in the tumor tissues compared to the normal tissues (Figure 8). In addition, TRIB3 expression is even higher in metastatic lesions compared to primary tumors.

Finally, evidence in the literature for gastric, liver, colorectal, renal cell, and non-small-cell lung carcinoma suggest increased TRIB3 protein level in the tumor tissue compared to the adjacent non-tumor tissue [14]. Taken together, these data indicate that in cancer, TRIB3 is upregulated both at mRNA and protein level.

### 6.2. TRIB3 Expression Levels and Patient’s Prognosis

Clinical studies report an increased TRIB3 expression in tumor tissue as compared to normal tissue, and this correlates with poor prognosis. In fact, colorectal cancer patients with high TRIB3 protein and mRNA expression levels are likely to experience a recurrence of the disease and display poorer overall survival [7]. In the same way, in renal cell carcinoma [8], gastric cancer [9], oral tongue squamous cell carcinoma [10], and non-small-cell lung carcinoma [11], the upregulated TRIB3 expression correlates with tumor stage, lymph node metastasis, disease recurrence, and thus with unfavorable prognosis. Nevertheless, despite the association of upregulation with poor prognosis, TRIB3 can act as a tumor-suppressor or oncogene in a context-dependent manner. However, no evidence could currently predict in which scenario TRIB3 would promote or inhibit cancer progression.

### 6.3. Is TRIB3 A Tumor Suppressor?

In some cellular and animal models of cancer, TRIB3 repression or deletion leads to enhanced cell proliferation and tumor formation in xenograft models, suggesting a potential role as tumor suppressor for TRIB3. The principal mechanism by which TRIB3 inhibits tumor progression probably relies on the inhibition of the phosphorylation/activation of Akt resulting in reduced pro-survival action [60,61]. Another mechanism by which TRIB3 can contribute to tumor suppression is via the interaction with the anti-viral cytidine deaminase APOBEC3A (Apolipoprotein B mRNA Editing Enzyme Catalytic subunit 3A). In cancer cells, two members from this family APOBEC3A and APOBEC3B are prevalent, act as potent mutators, and are associated with the burden of signature C to T mutations [62]. TRIB3 has been reported to restrict APOBEC3A activity by promoting its degradation via a proteasome-independent manner [63]. Such a TRIB3-APOBEC3A complex proposes TRIB3 as an important guardian of the genome integrity.

### 6.4. Is TRIB3 An Oncogene?

TRIB3 can contribute to tumor progression via the attenuation of the major intracellular degradative process named autophagy that has a potential preventive role against early stage cancer [64]. In an elegant study, Hua et al. reported that TRIB3 interacts with the autophagosome cargo protein p62, and restricts its co-aggregation with selected substrates tagged for degradation [65]. Thereby, the p62/TRIB3 2 interaction induces the blocking of the autophagic flux that interestingly leads to the accumulation of several tumor-promoting factors. The interruption of this p62/TRIB3 interaction attenuates xenograft tumor growth and metastatic dissemination, confirming the role of TRIB3 in this context as an oncogene [65,66].

In addition, TRIB3 can contribute to tumor progression via the positive regulation of signaling pathways like Notch and TGF-β. TRIB3 was identified as a master regulator of JAG1 (Jagged canonical Notch ligand 1) gene expression, and Notch activation in breast cancer. It induces cell survival and tumor xenograft growth [41,67]. The same authors demonstrated that MAPK-ERK was the predominant pathway promoting TRIB3-dependent Notch activation, and TGFβ-SMAD4 signaling in the regulation of JAG1 expression [41,67]. Similarly, in aggressive lung cancer cell lines, TRIB3 depletion is correlated with Notch1 downregulation and inhibition of tumor growth and metastatic dissemination [11]. In hepatocellular carcinoma cell lines, TRIB3 overexpression can increase the activity of TGFβ-SMAD3 by reducing the degradation of SMAD3 and promoting its translocation to the nucleus, which in turn promotes epithelial–mesenchymal transition [68].

## 7. Conclusions

Tribbles pseudokinase 3 clearly has a role in tumor progression, either as tumor suppressor or as oncogene. Nevertheless, in the wake of the impaired catalytic activity of its kinase-like domain, TRIB3 acts as a collaborating effector and modulates the localization or activity of other intracellular molecules. TRIB3 acts like a “jack of all trades” through a wide interaction network including components of central signaling pathways, such as Akt or MAPK, for example. In parallel, TRIB3 also interferes with gene expression at the transcriptional level, through direct interaction with various transcription factors or through impairment of the spliceosome machinery and pre-mRNA maturation. These complex interactive functions of TRIB3 are encouraging to better understand the processes in which it is involved, but also to explore novel cancer therapeutic approaches that target TRIB3.

## Figures and Tables

**Figure 1 cancers-13-01822-f001:**
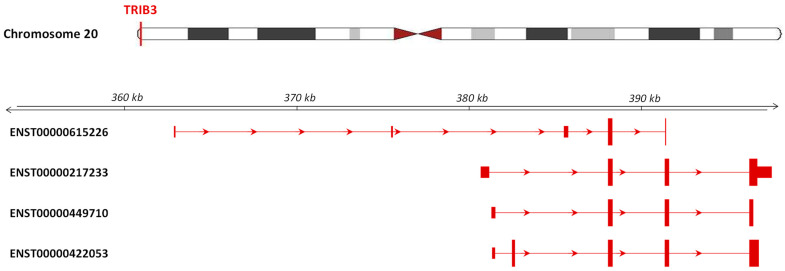
Transcript variant analysis of the *TRIB3* gene (ENSG00000101255.11). (Adapted from the Ensembl platform; https://www.ensembl.org accessed on 6 January 2021).

**Figure 2 cancers-13-01822-f002:**
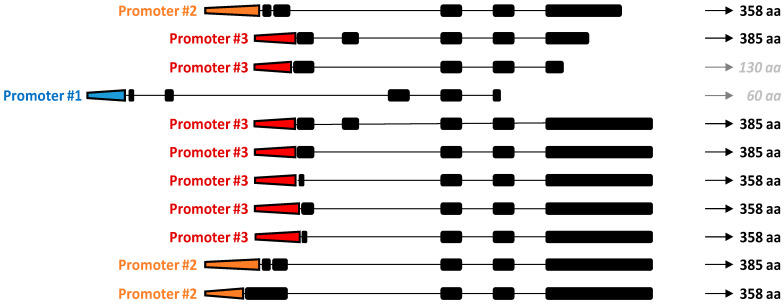
In silico analysis of the human *TRIB3* promoter region. The three putative alternative promoters and the related TRIB3 isoforms are indicated (Adapted from [12]). aa: amino acid.

**Figure 3 cancers-13-01822-f003:**
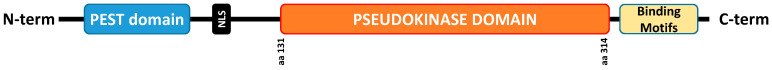
Functional domains of TRIB3 protein. NLS = nuclear localization signal.

**Figure 4 cancers-13-01822-f004:**
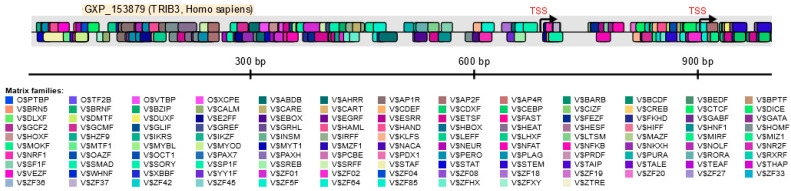
In silico analysis of the human *TRIB3* promoter region using the MatInspector Software (Genomatix Software GmbH, Munich; version v3.13). The matching transcription factor binding sites with a 5′ to 3′ (above the line) or a 3′ to 5′ (below the line) orientation relative to the transcription start site (TSS) are indicated.

**Figure 5 cancers-13-01822-f005:**
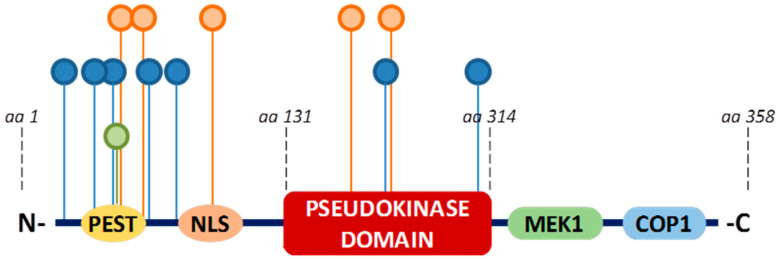
TRIB3 post-translational modifications. Phosphorylation (blue circle), ubiquitylation (orange circle), and methylation (green circle) sites reported from www.phosphosite.org accessed on 18 June 2020).

**Figure 6 cancers-13-01822-f006:**
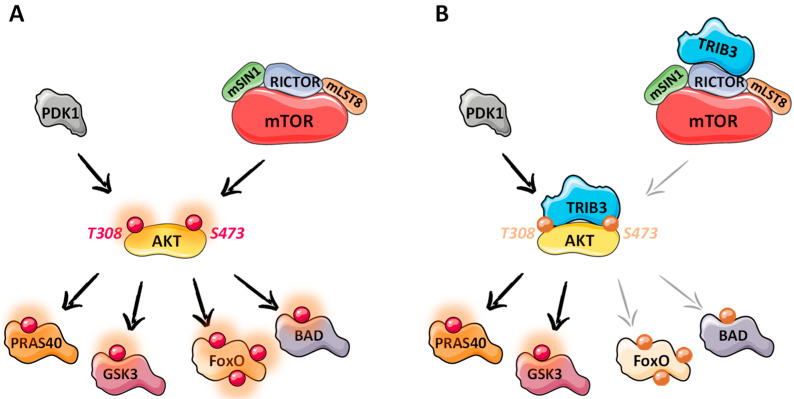
Putative mechanisms of TRIB3-mediated AKT signaling inhibition. Red circle = phosphorylation. (**A**) Activated signaling pathway in absence of TRIB3. (**B**) Partial inhibition of the signaling pathway in the presence of TRIB3.

**Figure 7 cancers-13-01822-f007:**
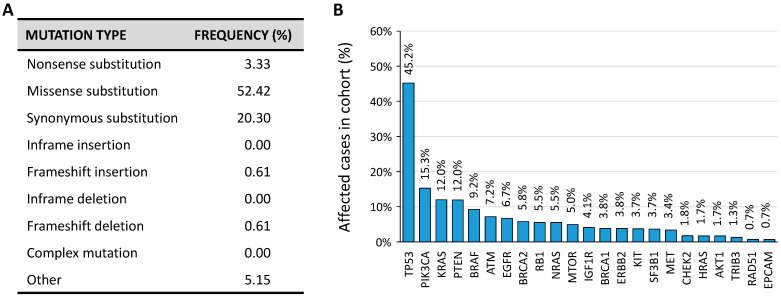
(**A**) Proportion of *TRIB3* somatic mutations according to the COSMIC database (Tate et al., 2019). (**B**) Non-exhaustive distribution of most frequently mutated genes according to the TCGA Research Network (https://www.cancer.gov/tcga accessed on 6 January 2021).

**Figure 8 cancers-13-01822-f008:**
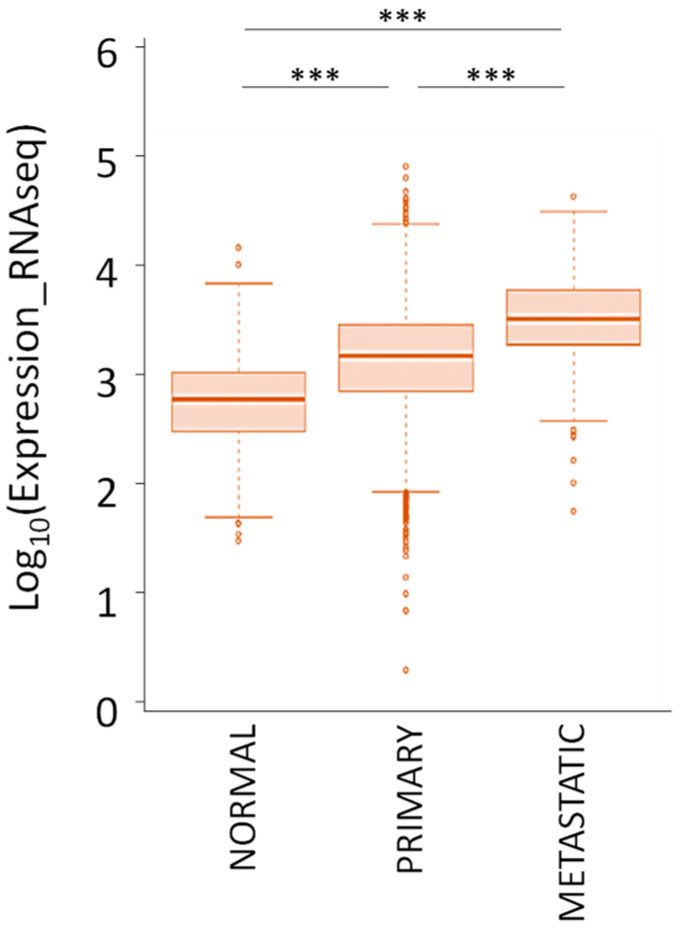
TRIB3 mRNA expression in normal and tumor tissues from 33 different cancer types according to the TCGA Research Network (https://www.cancer.gov/tcga accessed on 18 June 2020). *** *p*-value < 0.001.

**Table 1 cancers-13-01822-t001:** *TRIB3* Transcripts from Ensembl genome browser 101 (www.ensembl.org accessed on 6 January 2021).

Name	Transcript ID	bp	Protein	UniProt	Flags
TRIB3-201	ENST00000217233.8	2095	358aa	Q96RU7	TSL:1, APPRIS P3
TRIB3-202	ENST00000422053.3	1533	385aa	J3KR25	TSL:2, APPRIS ALT2
TRIB3-203	ENST00000449710.5	1070	271aa	B0QYQ2	CDS 3′ incomplete, TSL:5
TRIB3-204	ENST00000615226.4	877	130aa	A0A087WTX3	CDS 3′ incomplete, TSL:3

TSL1: all splice junctions of the transcript are supported by at least one non-suspect mRNA; SL2: the best supporting mRNA is flagged as suspect or the support is from multiple ESTs; TSL3: the only support is from a single EST; TSL5: no single transcript supports the model structure. APPRIS is a system to annotate alternatively spliced transcripts based on a range of computational methods. APPRIS P3: no clear principal variant identified, and more than one of the variants have distinct CCDS identifiers, APPRIS selects the variant with lowest CCDS identifier as the principal variant. APPRIS ALT2: candidate transcript that appears to be conserved in fewer than three tested species.

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
