# Peer review of "Tribbles Pseudokinase 3 Regulation and Contribution to Cancer"

_cancers, 2021, doi:10.3390/cancers13081822_

Round 1

Reviewer 1 Report

The manuscript by Stanovska et al competently reviews the regulation of Trib3 with a focus on its role in cancer. Focusing on its transcriptional and post-transcriptional regulation, Trib3’s role as a “scaffolding” protein in manifold pathways and a section on trib3 and cancer, the authors keep the article quite accessible and legible. As much as I see, the relevant literature is covered and properly presented.

A few minor issues should be resolved prior to publication.

- l35.  Trbl deficient  Drosophila embryos precociously go into mitosis but do not overproliferate in the mesoderm anlage.

- Fig. 4  higher resolution for the graphical items

- Fig. 6  please use an image in high resolution or better a vector-based illustration

- l 306 and fig 7. The statement for Trib3 and cancer is confusing. Why is “only” used as an adverb in l306. As even 1.3% are detected, it represents a high number. This statement is followed by the conclusion that there is no clear evidence for a function of Trib in cancer. For me this sounds contradictory.

- L 371  Do not start a sentence with “Because of …”

Author Response

Comments and Suggestions for Authors

The manuscript by Stefanovska et al competently reviews the regulation of Trib3 with a focus on its role in cancer. Focusing on its transcriptional and post-transcriptional regulation, Trib3’s role as a “scaffolding” protein in manifold pathways and a section on trib3 and cancer, the authors keep the article quite accessible and legible. As much as I see, the relevant literature is covered and properly presented.

A few minor issues should be resolved prior to publication.

- l35.  Trbl deficient  Drosophila embryos precociously go into mitosis but do not overproliferate in the mesoderm anlage.

>> We agree with the Referee and rectified the sentence as follow “Tribbles-deficient embryos enter mitosis early in the mesodermal cells that impairs morphogenetic movement of gastrulation…”.

- Fig. 4  higher resolution for the graphical items

>> We adjusted the resolution for the figure 4 in the revised version of the manuscript.

- Fig. 6  please use an image in high resolution or better a vector-based illustration

>> We adjusted the resolution for the figure 6 in the revised version of the manuscript.

- l 306 and fig 7. The statement for Trib3 and cancer is confusing. Why is “only” used as an adverb in l306. As even 1.3% are detected, it represents a high number. This statement is followed by the conclusion that there is no clear evidence for a function of Trib in cancer. For me this sounds contradictory.

>> We removed the adverb “only” in this sentence in the revised version of the manuscript. The TRIB3 mutation level is indeed in a range similar to HRAS, AKT1 or RAD51 that are key elements in cancer, but there is no evidence about the relevance of somatic TRIB3 mutations in cancer. In terms of correlation between TRIB3 alteration and diseases, the variation in the expression level has a greater impact than mutational status.

- L 371  Do not start a sentence with “Because of …”

>> We corrected the sentence in the revised version of the manuscript.

>> We thank very much this Referee for his/her constructive comments that greatly improved the manuscript.

Reviewer 2 Report

A comprehensive summary of TRIB3 function, regulation, and its complex role in cancer. There are some error/omissions that should be corrected.

  1. Figure 3, error in domain structure, the N, C termini label is wrong.
  2. Incorrect or missing reference
  • Line 44, ref. 5 does not contain information as written in the text.
  • Line 286, needs reference
  • Line 296-7 needs reference
  1. Line 209, should be MEK1 (binding) domain.

The reviewer would like to suggest the author elaborate more on the following points.

  • A brief summary of the three TRIB gene/proteins and how is TRIB3 unique.
  • A few sentences on what is the functional consequence intron retention.
  • It is clear from the detailed summary of literature that the role of TRIB3 in cell signaling and cancer is complicated and context dependent. It would be tremendously beneficial if the author can elaborate more on the seemingly contradictory roles of TRIB3. Are there any general rules/principles that underlies when, where, and how TRIB3 exhort opposite functions?

Author Response

A comprehensive summary of TRIB3 function, regulation, and its complex role in cancer. There are some error/omissions that should be corrected.

  1. Figure 3, error in domain structure, the N, C termini label is wrong.

>> We apologize for this mistake, and corrected the figure 3 in the revised version of the manuscript.

  1. Incorrect or missing reference
  • Line 44, ref. 5 does not contain information as written in the text.

>> We apologize for this mistake, and modified the reference to cite the expected publication by Kiss-Toth et al, 2004 in the revised version of the manuscript.

  • Line 286, needs reference

>> We added the citation of Stefanovska et al, 2020 in the revised version of the manuscript.

  • Line 296-7 needs reference

>> We added the citation of Prudente et al, 2012 in the revised version of the manuscript.

  1. Line 209, should be MEK1 (binding) domain.

>> We apologize for this omission, and we corrected it in the revised version of the manuscript.

The reviewer would like to suggest the author elaborate more on the following points.

  • A brief summary of the three TRIB gene/proteins and how is TRIB3 unique.

>> We now mentioned the discrepancies and similarities between TRIBs homologs in the introduction section, and referred to the comprehensive review by Eyers et al, 2016.

  • A few sentences on what is the functional consequence intron retention.

>> We mentioned the functional consequence of IR in the paragraph 5.3 of the revised manuscript.

  • It is clear from the detailed summary of literature that the role of TRIB3 in cell signaling and cancer is complicated and context dependent. It would be tremendously beneficial if the author can elaborate more on the seemingly contradictory roles of TRIB3. Are there any general rules/principles that underlies when, where, and how TRIB3 exhort opposite functions?

>> As stated, TRIB3 can act as a tumor-suppressor or oncogene in a context-dependent manner. However, no evidence could currently predict in which scenario TRIB3 would promote or inhibit cancer progression. This has been added in the revised version of the manuscript.

>> We thank very much this Referee for his/her constructive comments that led to improve the manuscript.

Reviewer 3 Report

Overall, a good and comprehensive review of TRIB3 in the context of cancer.

Minor grammatical points:

Line 108: 1/ change to 1)

Line 110: 2/ change to 2)

Line 128: ‘Evidences’ to ‘evidence’

Line 178: ‘up to date’ change to ‘to date’

Figure 4: Slightly pixelated, increase resolution of figure if possible.

Author Response

Overall, a good and comprehensive review of TRIB3 in the context of cancer.

Minor grammatical points:

Line 108: 1/ change to 1)

Line 110: 2/ change to 2)

Line 128: ‘Evidences’ to ‘evidence’

Line 178: ‘up to date’ change to ‘to date’

>> We corrected all these typos in the revised version of the manuscript.

Figure 4: Slightly pixelated, increase resolution of figure if possible.

>> We adjusted the resolution of the figure 4 in the revised version of the manuscript.

>> We thank very much this Referee for his/her constructive comments that led to improve the manuscript.